# Human iPSCs as Model Systems for BMP-Related Rare Diseases

**DOI:** 10.3390/cells12172200

**Published:** 2023-09-02

**Authors:** Gonzalo Sánchez-Duffhues, Christian Hiepen

**Affiliations:** 1Nanomaterials and Nanotechnology Research Center (CINN-CSIC), ISPA-HUCA, Avda. de Roma, s/n, 33011 Oviedo, Spain; 2Department of Cell and Chemical Biology, Leiden University Medical Center, Einthovenweg 20, 2333 ZC Leiden, The Netherlands; 3Department of Engineering and Natural Sciences, Westphalian University of Applied Sciences, August-Schmidt-Ring 10, 45665 Recklinghausen, Germany

**Keywords:** PAH, HHT, FOP, FAP, TGF-beta

## Abstract

Disturbances in bone morphogenetic protein (BMP) signalling contribute to onset and development of a number of rare genetic diseases, including Fibrodysplasia ossificans progressiva (FOP), Pulmonary arterial hypertension (PAH), and Hereditary haemorrhagic telangiectasia (HHT). After decades of animal research to build a solid foundation in understanding the underlying molecular mechanisms, the progressive implementation of iPSC-based patient-derived models will improve drug development by addressing drug efficacy, specificity, and toxicity in a complex humanized environment. We will review the current state of literature on iPSC-derived model systems in this field, with special emphasis on the access to patient source material and the complications that may come with it. Given the essential role of BMPs during embryonic development and stem cell differentiation, gain- or loss-of-function mutations in the BMP signalling pathway may compromise iPSC generation, maintenance, and differentiation procedures. This review highlights the need for careful optimization of the protocols used. Finally, we will discuss recent developments towards complex in vitro culture models aiming to resemble specific tissue microenvironments with multi-faceted cellular inputs, such as cell mechanics and ECM together with organoids, organ-on-chip, and microfluidic technologies.

## 1. Introduction:

### 1.1. Use of iPSCs for Efficient BMP Rare Disease Research and Orphan Drug Discovery 

The study of rare diseases associated with mutations in the bone morphogenetic protein (BMP) signalling pathway has been advanced by the advent of human induced pluripotent stem cell (hiPSC) technology through Sir John B. Gurdon, Shinya Yamanaka, and co-workers in 2006 [1], reviewed in [2]. The definition of “rare” or “orphan” disease depends on the jurisdiction of every country. A condition that affects 1 in 2500 individuals may be deemed “rare” based on a global average [3]. Genetic factors account for more than 80% of rare diseases [4]. Given the low numbers of individuals affected, which limits the commercial impact of a marketable drug, the pharmaceutical industry may have less interest in the study of rare diseases. To compensate for this, the pharmaceutical authorities often implement measures to attract the industry into the field of orphan diseases, for example, by accelerating the bureaucratic procedures, offering longer patent licenses, and reducing the number of individuals required for clinical trials. However, the status quo is more complex. Learning from rare diseases, pathogenic mechanisms may be identified that can be applied to more prevalent conditions. The BMP-related rare genetic disease (herein referred to as BMP rare disease) Fibrodysplasia ossificans progressiva (FOP) is one such example. In FOP, genetic mutations in the *ACVR1* gene (ALK2) (described later in more detail) induce extraskeletal bone formation [5,6]. Mechanistically, the mutant ALK2 causes a stimuli-dependent over-activation of the BMP pathway, for which small molecule inhibitors have been developed. Studies with such compounds have revealed that systemic BMP inhibition may lead to iron overload. Therefore, some of these molecules, which may not be ultimately adequate for FOP, are repositioned into more common disorders like anaemia (clinical trial NCT05090891). Two related rare vascular diseases that are currently being investigated by iPSC technology and which will be covered here are Pulmonary arterial hypertension (PAH) and Hereditary haemorrhagic telangiectasia (HHT), also known as Morbus Osler–Weber–Rendu disease. A deeper understanding of PAH and HHT promises benefits for other, more prevalent (cardio)vascular diseases with related biological features. 

Throughout the drug development pipeline, about 90% of the screened compounds usually drop out before the pre-clinical phase is initiated [7]. Even after entering pre-clinical testing, promising candidates may be discarded because results from in vitro experiments cannot be reproduced in animals. As we will discuss later in the cases of FOP and PAH, disease animal models do not necessarily resemble the entire human patient pathophysiology [8]. Therefore, drug mechanisms discovered in animals may translate poorly to patients during clinical trials, dampening the success rate of a therapeutic compound and its ability to reach the market. In order to facilitate the transition from preclinical research to clinical trials, it is crucial to develop and implement appropriate humanized cellular systems for evaluating drug efficacy, toxicity, and pharmacokinetics. Human derived iPSCs are superior to human primary cells or cell lines with unlimited laboratory life-span if iPSCs are generated, cultured, and differentiated under the right conditions. Applying an organoid culture of human-iPSC-derived cell types with complex environmental cues can mimic major aspects of the healthy and diseased human body [9]. These additional cues may include, for example, biomechanical stimulation, hypoxic and/or inflammatory microenvironments, and the right co-culture combination of cell types involved in the biological process. Ideally, these platforms will allow researchers to interrogate the complexity of BMP signalling, and identify novel molecules with potential as druggable targets and/or biomarkers in rare diseases. 

### 1.2. Basics in BMP Signalling

Following their discovery by Marshall Urist as ectopic bone inducing agents in rodents and their subsequent isolation and characterization [10,11,12], the BMPs were included as members of the transforming growth factor (TGF)-β superfamily, due to their high sequence and structural homology with existing TGF-β members. The TGF-β family of cytokines consists of soluble growth and differentiation factors, which exert pleiotropic functions through the activation of specific heterotetrameric receptor complexes on the cell membrane (Figure 1). The BMP receptors exhibit serine/threonine kinase activity, transducing extracellular signals into intracellular signalling networks to regulate, e.g., gene expression. During the past decades, comprehensive studies have emphasized the complexity of the TGF-β signalling pathway at different levels. At the extracellular level, over 30 different ligands, membrane and soluble receptors, natural antagonists, and extracellular matrix (ECM) proteins interact and define the activation of different subsets of TGF-β receptors (TGFβRs). Intracellularly, the different branches of the pathway tend to counteract one another and crosstalk to a myriad of other signalling pathways. As a result, complex intertwined transcriptional networks are regulated, in cooperation with tissue- and stimulus-specific transcriptional (co)-factors.

Canonical BMP signalling is initiated upon the interaction between soluble dimeric BMP ligands and so-called BMP type I receptors (BMPRI) (Figure 1). Four BMPRI, also termed activin-like-kinase (ALK) receptors, have been described: ALK1, 2, 3, and 6. The different BMP ligands display distinct affinities for the ALK receptors (i.e., BMP9/10 are high affinity ALK1 binders, BMP-2/-4 preferably bind ALK3/6, and BMP-5/-6/-7 preferably bind ALK2, reviewed in [13,14]), but the interaction is determined by the expression levels and availability of ligands and receptors. As such, soluble and membrane-bound factors like the co-receptors endoglin (*ENG*) and Cripto (*TDGF1*) can enhance BMP-9/-10-induced interaction with ALK1 [15,16]. These co-receptors can be shed off the membrane and circulate in the extracellular space [15,17,18]. The repulsive guidance molecules (RGMs) can either hamper or enhance ligand–receptor interaction [19]. BAMBI (BMP and activin membrane-bound inhibitor) is a transmembrane protein able to bind extracellular ligands, but it lacks the intracellular BMP type I receptor kinase domain [20]. Neogenin can bind and inhibit signalling induced by BMP-2, -4, -6, and -7 [21]. The protein casein kinase II (CK2) appears to be associated with ALK3 in C2C12 cells and is released upon stimulation with recombinant BMP2. Genetic and pharmacological inhibition of membrane casein kinase II (CK2) augmented BMP2-induced transcriptional activity, suggesting a functional role for CK2 on BMP signalling [22]. Similarly, an siRNA-based screening using a luciferase reporter line identified the membrane proteins fragile histidine triad (FHIT) [23], lymphocyte cell-specific protein tyrosine kinase (LCK), Src-like Kinase (FYN) [24], and protein tyrosine phosphatase non-receptor type 1 (PTNP1) [25] as contributors to BMP transcriptional responses. Cysteine-rich motor neuron 1 protein (CRIM1) is a glycosylated transmembrane protein that interferes with BMP ligand maturation and extracellular expression [26]. Furthermore, BMP signalling is also regulated through interaction of BMPs with the ECM, thereby restricting the local BMP bioavailability. BMPs interact, e.g., with fibronectin [27,28], collagens [29,30], fibrillin [31,32], and heparins [33]. Heparin binding of some BMPs (including BMP-2) was shown to have dual activity. In some contexts, heparin binding represses [34,35], while, in other contexts, it increases the BMP signalling output [36,37]. Finally, there are soluble BMP antagonists like noggin, chordin, the gremlins, and the follistatins that can sequester and shield the ligands, in order to avoid their interaction with the extracellular domain of the receptors [38]. 

The association between the ligand dimer and a type I receptor dimer favours the recruitment of the type II receptors, thereby forming a heterotetrameric receptor complex (Figure 1). There are three types of BMP type II kinase receptors, named ActRIIA, ActRIIB, and BMPR2 (Figure 1). Unlike the type I receptors, the BMP type II receptors are constitutively active and their ligand-induced incorporation into the receptor complex leads to the trans-phosphorylation of the type I receptors in their cytosolic Glycine Serine (GS) domain. This phosphorylation provokes a conformational change within the cytosolic domain of the BMPRI and displaces the intracellular receptor inhibitor FK506 binding protein 1a (FKBP12). This, in turn, allows for the interaction of the cytosolic kinase domain with the BMP intracellular effectors, the different “mothers against decapentaplegic homolog” (SMAD) proteins. Upon interaction with the BMPRI-kinase domain, the receptor-regulated SMADS (R-SMADS) 1/5/8 (SMAD8, also known as SMAD9) are transphosphorylated, and thereby activated, in their C-terminal domain. The activation of the TGF-β type I receptors (ALK4/5/7) and their downstream signalling pathway, including the phosphorylation of a different subset of R-SMADS, SMADS 2/3, exhibit some peculiarities. Due to the space limitation, we will not further expound on TGF-β specific regulation, signalling, crosstalk, and particular target genes and, instead, recommend reading [39]. Inhibition of TGFβR internalization hampers R-SMADS phosphorylation. Two major proteins are responsible for the endocytosis-mediated internalization of ligand-bound membrane receptors. While SARA [40] mediates ALK4/5/7 endocytosis, endofin negatively modulates BMP signals through receptor dephosphorylation [41].

Once phosphorylated, the R-SMADS interact with a common SMAD (Co-SMAD or SMAD4) to form a heterotrimer that translocates into the nucleus (Figure 1). SMAD4 is shared by both BMP and TGF-β R-SMADs; therefore, the amount of free SMAD4 molecules may limit R-SMAD signalling. Once in the nucleus, SMAD complexes associate with further DNA binding (co-)factors to regulate the expression of specific genes. Classical BMP target genes are the differentiation and proliferation associated inhibitor-of-differentiation (ID) factors (*ID1*, *ID2* and *ID3*) or genes involved in osteogenic differentiation (i.e., *SP7/Osterix*, *RUNX2*). *SMAD6* is also a BMP signalling target gene. SMAD6 and SMAD7 are inhibitory SMADs (I-SMADs), which serve as negative feedback loops in the pathway. SMAD6/7 facilitate the proteosomal-mediated degradation of BMPR, and also compete with the R-SMAD for BMPR binding [42,43,44,45].

In addition to SMAD activation, BMPRs also modulate the activation of so-called non-canonical pathways (Figure 1). These are less well characterized and involve different branches of mitogen activated protein kinases (MAPK) pathways, like p38, the extracellular signal-regulated kinase (ERK), c-Jun N-terminal Kinase (JNK), phosphoinositide-3-kinase (PI3K), mammalian target of rapamycin (mTOR), glycogen synthase kinase 3 beta (GSK3-β), and small Rho GTPases in the Rho/Rac subfamily such as CDC42, as well as the LIMK/cofilin pathway, amongst others [46,47,48] (Figure 1). It was shown that, e.g., activation of BMP non-canonical pathways can result in direct non-transcriptional responses of mesenchymal progenitors, e.g., those required for cytoskeletal rearrangements and cell migration [49]. Unlike SMAD signalling, which is strong and immediate after BMPR activation, non-canonical pathways are often weakly induced by BMP ligands, and the activity can be masked by other environmental cues (i.e., inflammatory cytokines leading to inflammatory nuclear factor (NF)-kB signalling, fibroblast growth factor (FGF) signalling, vascular endothelial growth factor (VEGF) signalling, Wnt signalling, or hypoxia-HIF1α signalling). Importantly, the transcriptional output of SMAD activation can be fine-tuned by non-canonical BMPR pathways and through further crosstalk with other signalling cascades [50,51]. As such, MAPK, ERK, and JNK can crosstalk to the canonical SMAD signalling by directly phosphorylating SMADs in the linker region, thereby competing with the phosphorylation at the C-terminus domain by the BMPRs. Moreover, many other input-dependent transcription factors partner with SMAD complexes, thereby determining the binding affinity and specificity of SMAD complexes to certain target genes [52]. Therefore, the effect of BMP ligands may be dependent on multifaceted cues and factors that eventually trigger a cell-type or organ-specific response.
Figure 1**BMP signal transduction**. The interaction between BMP ligands and BMP type I receptors induces the formation of membrane heterotetrameric complexes. This interaction is influenced by the relative availability of ligands and receptors, which is regulated by soluble antagonists, co-receptors, and extracellular matrix (ECM) components. The activated BMP receptors transmit signals intracellularly through their serine/threonine kinase domain, leading to the phosphorylation of the BMP canonical mediators, SMAD1/5/9, at the C-terminal domain. Moreover, BMP receptors regulate the activation of non-canonical signalling pathways, which serve as a signalling hub integrating tissue-specific environmental cues. Canonical and non-canonical signalling pathways intersect at various levels. Once activated, SMAD1/5/9 trimerize with SMAD4 and translocate into the nucleus to bind specific gene promoters in collaboration with transcription co-factors. A number of environmental cues co-influence signalling outcomes, including cellular mechanics such as mechanical shear forces or inflammatory or hypoxic microenvironments. Due to space limitations, details on the TGF-β-specific regulation of the signalling branch are not shown.
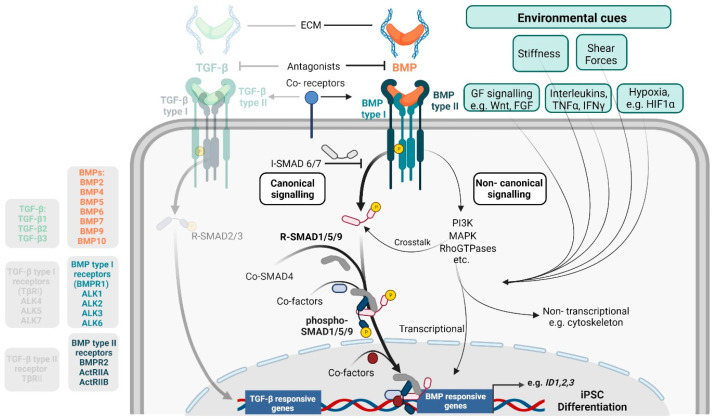


### 1.3. Cell-Type Specific BMP Activity

Physiologically, almost all BMP ligands have been found to circulate systemically at functional levels, and some BMP receptors are abundantly expressed in a plethora of cell types. However, the expression of some BMPs is restricted to specific tissues. For example, BMP-9 is mostly expressed in the liver [53], while BMP-10 expression is restricted to the right cardiac atrium in adulthood [54,55]. The localization of BMP-2 transcripts was observed in various areas, such as the developing heart, whisker follicles, tooth bud, and, as expected, in the limb bud. BMP-4 transcripts, on the other hand, were detected in limb buds, nervous tissue, and craniofacial tissues through in situ hybridization studies [56]. BMP-7 is mainly expressed in the kidney [57]. Once in circulation, BMP ligands show different affinity for BMPRs; therefore, the relative concentration of ligands and expression levels of the BMPRs can determine the activation of signalling. While ALK2 and ALK3 are broadly expressed, ALK1 expression is predominantly found in endothelial cells (ECs) and, at some level, in chondrocytes [16] and pulmonary vascular smooth muscle cells (VSMCs) [58]. ALK6 is highly expressed in ovary [59] and prostate tissue [60]. Moreover, BMP ligands are promiscuous and can bind different receptor complexes depending on their availability. BMP-9, for example, can bind to ALK1 and ALK2, although the affinity for ALK1 is much higher (EC50 = 50 pg/mL for ALK1 and 50 ng/mL for ALK2) [61]. The affinity can be severely influenced by type III receptors or co-receptors, such as betaglycan, which lack a cytosolic domain with enzymatic activity but can shield or present ligands to type I and type II receptors. The expression levels of co-receptors also differ among cell types. Endoglin is highly expressed in embryonic tissues and stem cell-like cells in adulthood, as well as in ECs and few subsets of immune cells. Furthermore, environmental cues, like inflammation, can influence the expression of BMPRs. As such, BMPR2 is inhibited at different levels by inflammatory interleukin (IL)-6 and tumour necrosis factor (TNF)-α signalling, for example [62,63,64], thereby disturbing BMP signalling under inflammatory microenvironmental conditions [63]. Hypoxia, for instance in highly condensed chondrocytes, extends the duration of SMAD1/5/9 activation. Mechanical forces also influence BMP activity (Figure 1). Shear stress can tune the endoglin co-receptor incorporation into BMPR complexes in ECs and sensitize them to low BMP-9 concentrations [65,66], and mechanical loading can promote the autocrine activity of ligands such as BMP-2 and influence BMP target gene regulation [58,67]. Moreover, culturing cells on a soft vs. a stiff matrix influences osteogenic BMP signalling, endocytosis of BMPRs, and incorporation of transcriptional co-factors into the SMAD signalling complex [68,69]. The special role of cell mechanics for the modelling of BMP rare diseases by iPSCs will be discussed in Section 3. 

Although the pathways by which BMP ligands induce downstream signal activation have been roughly characterized in the previous decades, it is only now, when using more complex human cell models, that tissue-specific determinants of BMP signalling are starting to be unveiled. This highlights the need for studying BMP signalling in the right cellular and microenvironmental context, resembling, when possible, the cellular, biochemical, and biomechanical conditions which drive the onset and development of BMP related human diseases.

### 1.4. BMPs in iPSC Stemness and Differentiation

During the early embryonic development of vertebrates, gradients of BMP ligands and inhibitors pattern the dorsal–ventral embryonic axis [70,71]. BMP signalling specifies the ventral mesoderm [72,73] and affects neural tube formation [74], intestines (reviewed in [75]), liver (reviewed in [76]), and kidney development [77]. Importantly, BMP signalling becomes the key to maintaining stemness in embryonic stem cells, germline stem cells, hematopoietic stem cells, and intestinal stem cells in a species-dependent manner (reviewed in [78]). In mouse embryonic stem cells (mESCs), BMPs control self-renewal and promote stemness as they prevent ectoderm differentiation [79,80]. In human ES cells (hESCs), BMPs also inhibit neural differentiation but are mostly known as potent inducers of mesoderm formation. The function of BMP signalling might be different in iPSCs, where BMP signals seem to not be required for the maintenance of human iPSC stemness, while Activin-A signalling is involved in maintenance of human iPSC pluripotency [81]. Yamanaka and Conklin have shown that BMP-SMAD-ID signalling promotes episomal vector mediated reprogramming of differentiated somatic cells to pluripotency, since adding BMP-4 during the early phase of reprogramming inhibited p16/INK4A-dependent senescence, a major barrier to efficient reprogramming [81] (Figure 2). Typically, the efficacy of iPSC reprogramming is quite low (less than 1% of human primary somatic cells that have received reprogramming factors turn into iPSCs) [82]. Upon successful reprogramming, most iPSC differentiation protocols describe short-term BMP treatment if mesoderm induction and differentiation of iPSCs towards a definitive endoderm is desired. For ectodermal differentiation, BMP activity needs to be strictly repressed initially, and only at later ectodermal specification, e.g., towards skin progenitors or keratinocytes, is BMP signalling again required [82,83] (Figure 2). The most used TGF-β superfamily ligands found in human iPSC mesodermal or definite endodermal differentiation protocols are Activin-A [84,85], BMP-4 [86,87,88], BMP-2 [89,90], and BMP-7 [91,92]. Few other established iPSC differentiation protocols investigated the potency of other, less common, BMP ligands such as BMP-10. The latter is particularly interesting since it can induce human iPSC differentiation towards extraembryonic trophoblast tissue [93]. It is important to note that concentrations of different BMP ligands can vary between published protocols, even when the targeted cell differentiation is the same. For instance, in iPSC-derived EC differentiation, which often includes BMP-4-dependent mesoderm induction, one can find a comparison of different protocols in [94]. Importantly, BMP stimulation alone is not sufficient to induce iPSC differentiation towards specific cell types. Therefore, BMPs are usually combined with other growth factors (e.g., FGF, Wnt, or VEGF) (Figure 1) and/or pathway inhibitors, such as small molecule inhibitors targeting TGF-β type I receptors ALK4/5/7 or Wnt/GSK3-β signalling (SB431542 and CHIR99021, respectively) (Figure 2). We will herein limit our review on the most important mesodermal-derived cell types derived from iPSCs, which are relevant for the BMP rare diseases FOP, PAH, and HHT. These include fibroblasts, macrophages, fibro-adipogenic progenitors (FAPs), muscle-residing satellite cells, skeletal myoblasts and myocytes, osteoblasts and osteocytes, chondroblasts and chondrocytes, and cell types forming the vasculature, including ECs and mural cells such as VSMCs and pericytes. 

## 2. iPSC Technology and BMP-Related Rare Diseases

BMP-related rare diseases affect a small number of individuals, and there are currently a limited number of clinical treatment strategies or interventions available. Mutations in genes encoding for BMP pathway proteins alter either their expression or protein function. Three conditions have garnered significant research attention in efforts to comprehend the genotype–phenotype relationship. These are Fibrodysplasia ossificans progressiva (FOP), Pulmonary arterial hypertension (PAH) and Hereditary haemorrhagic telangiectasia (HHT). However, other BMP rare diseases have been described. Diffuse intrinsic pontine glioma (DIPG) refers to a devastating form of brainstem glioma with highly proliferative and micro-vascularized solid tumours [95], often arising during childhood. These tumours are often inoperable and exhibit a very poor survival rate. Over 80% of tumours have K27M mutations in H3.3 (*H3F3A*) and H3.1 (*HIST1H3B*) genes, leading to reduced H3K27 trimethylation. DIPG exhibits heterogeneity due to additional somatic mutations, including *TP53* inactivation in 40–50% of patients. Notably, *ACVR1* (ALK2) somatic mutations occur in approximately 33% of DIPGs, sharing similarities with those FOP mutations described below [96,97]. Three additional mutations (L343P, R307L, and H286N) in the cytosolic domain of ALK2 have been linked with human diseases. While L343P and R307L were linked to defects in the atrioventricular septum, H286N was found in one patient with Down syndrome and congenital heart disease (CHD). CHD was also associated with several loss of function mutations in *BMPR1A* (ALK3) [98,99]. Germline mutations in *ALK3* leading to truncated protein receptors have been traditionally associated with juvenile polyposis syndrome (JPS), which is an autosomal dominant gastrointestinal disorder that predisposes for gastrointestinal cancers [100]. Previously, JPS was linked to mutations in *MADH4* (encoding SMAD4) [101]. Finally, inactivating mutations in *BMPR1B* (ALK6) result in uncommon inherited skeletal disorders, named acromesomelic dysplasias (AMD), which are characterized by short stature, extremely short limbs, and deformities in the hands and feet [102,103]. To the best of our knowledge, iPSC technology has been applied so far to the fields of FOP, PAH, and HHT research (Figure 2). Therefore, in the subsequent paragraphs, we will summarize the implementation of human iPSCs as model systems to investigate these three BMP rare diseases. 

### 2.1. Fibrodysplasia Ossificans Progressiva and hiPSCs

Heterozygous germline gene mutations in the *ACVR1* gene were found in all patients diagnosed with Fibrodysplasia ossificans progressiva (FOP, OMIM 135000) [104], while approximately 97% of all patients share the exact gene point mutation c.617G>A; (p.R206H) in the chromosome 2 (Table 1). The gene *ACVR1* encodes for the protein ALK2, a BMP type I receptor, which classically mediates signal transduction of osteogenic BMP ligands like BMP-5, BMP-6, and BMP-7 [13] (Figure 3). Despite the fact that ALK2 is broadly expressed, the main clinical manifestation of FOP is limited to connective tissues within ligaments, fascia, tendons, and joints. Affected individuals are recognized by congenital skeletal malformations (i.e., great toe deformity or *hallux valgus*) and postnatal episodic heterotopic ossification (HO) (Figure 3). During the first decade of life, sporadic episodes of painful soft tissue swellings (“flare-ups”) occur often following soft tissue injury, intramuscular injections, viral infection, muscular stretching, falls, or fatigue [105]. These events usually result in the formation of extraskeletal bone plaques of endochondral origin caused by the differentiation of existing fibroadipogenic progenitor cells (FAPs) into cartilage first, and, eventually, into mature bone [106]. Other, often overlooked, features of FOP include cardiac abnormalities in new-borns, such as ventricular septal hypertrophy and cardiac conduction abnormalities in children [107,108,109,110,111,112]. The blood vessels in FOP lesions are over-represented in comparison to non-genetic forms of HO, exhibit weak junctions, and are prone to oedema [110,113], suggesting additional effects of ALK2^R206H^ in cell types other than bone progenitor cells.

The first reported FOP iPSC line was established upon genetic reprogramming of skin fibroblasts using Sendai Virus (SeV) particles carrying *OCT3/4*, *SOX2*, *KLF4*, and *MYC*, and maintained on mitomycin-C-treated mouse embryonic fibroblast (MEF) feeder cells [114] (Table 1). Remarkably, the authors found a substantial drop in reprogramming efficiency in FOP iPSCs from donors carrying the ALK2 mutations R206H and G356D when compared with non-isogenic control cells. This finding was unexpected, as it contradicted the previously mentioned observations of enhanced reprogramming efficacy following BMP-SMAD-ID activation [82]. Here, upon incubation with the ALK2-biased small molecule kinase inhibitor LDN-193189, the efficiency was increased in a dose-dependent manner (Figure 2), suggesting that the kinase activity of ALK2 may compromise pluripotent transformation. Congruently, the expression of mesodermal and endodermal gene markers was upregulated in FOP iPSC not treated with LDN-193189, whereas the expression of the neuroectodermal markers *SOX1* and *NESTIN* remained unaffected. In conclusion, the generation of iPSCs from FOP patients’ somatic cells may require inhibition of mutant ALK2 activity by adding small molecule inhibitors, such as ALK2 inhibitors like LDN-193189 or dorsomorphin to maintain pluripotency (Figure 2). In 2013, Matsumoto et al. described the generation of iPSCs from human dermal fibroblasts from five FOP donors and six control donors using two different reprogramming methods [115] (Table 1). The authors explored the use of retrovirus and episomal vectors carrying *OCT4*, *SOX2*, *KLF4*, and *MYC* to generate iPSCs maintained on Matrigel or feeder cells. Unlike the previous report, Matsumoto et al. did not find any differential efficiency in the generation of iPSCs from FOP donors compared to control cells, and ALK2 kinase inhibition was not required for a successful reprogramming. Interestingly, FOP iPSCs demonstrated augmented mineralization and chondrogenesis in vitro, a phenomenon that could be mitigated in the presence of LDN-193189. Cai et al., reprogrammed urine renal epithelial cells from control and FOP individuals, cultured in serum-free medium supplemented with small molecules CHIR-99021, PD-0325901, A83-01, and thiazovivin. Transient expression of *OCT4*, *SOX2*, *KLF4*, and the pCEP4-miR-302-367 cluster was achieved through electroporation with episomal vectors [116]. In this case, the reprogramming of FOP iPSCs was also not less efficient than in control cells. Moreover, the authors followed established protocols to differentiate control and FOP iPSCs into pericytes and ECs [117,118] (Table 1). Interestingly, FOP derived ECs were less proliferative and stable, and exhibited lower VEGFR2 expression compared with non-isogenic control cells. The expression of endothelial-specific markers was reduced, whilst fibroblast genes were upregulated in FOP cells, suggesting that FOP ECs were prone to undergo a developmental dedifferentiation process termed endothelial-to-mesenchymal transition (EndMT) [119]. Through EndMT, ECs progressively transform into multipotent MSC-like cells and may give rise to osteogenic and chondrogenic cells, at least in vitro, consistent with previous findings [112,120] (Figure 2). iECs were also investigated by Hsiao’s lab, using non-isogenic previously published iPSC lines [115] (Table 1). Unlike the previous method, in this case, EC differentiation was achieved through a protocol not including the TGF-β receptor small molecule inhibitor [121]. The expression of the EC marker VE-Cadherin (*CDH5*) was significantly low, whereas the mesenchymal genes *FSP-1*, collagen 2-alpha-1 (*COL2A1*), *COL1A1*, and *ALPL* were consistently elevated in FOP iECs. Despite the fact that this gene expression pattern might be suggestive of EndMT, the authors failed to obtain functional osteoblast-like mineralizing cells in vitro, either from control or from FOP iECs. Lineage tracing studies in mouse models of FOP have pointed at a population of FAP cells defined as Tie2^+^, CD31^−^, CD45^−^, PDGFRα^+^, SCA1^+^ as the independent mediator of HO in FOP [122]. FAPs may be responsible for both congenital and postnatal bone abnormalities in patients, but ubiquitous gene expression of *ALK2^R206H^* is embryonically lethal in mice [122,123,124]. Therefore, iPSC technology has been applied to FOP in order to model potential HO progenitor cells in FOP (Figure 3). In the elegant work by Nakajima et al. [125], the authors explore the chondrogenic potential of different somite derivatives obtained from isogenic control and FOP iPSCs. FOP iPSC-derived pre-somites differentiated towards MSCs were able to form chondrogenic cartilage-like structures in vitro in response to Activin-A. However, differentiation towards sclerotome resulted in cells that could successfully form chondrogenic 3D pellets in vitro, although no neo-functional response to Activin was observed in FOP cells. In FOP mice, however, tendon resident scleraxis (*Scx*) positive cells have been proposed as Activin-responsive sporadic bone progenitor cells [124]. This might suggest that embryonic ALK2^R206H^ expression may impact differently the cell fate of mouse and human bone progenitor cells. Muscle regeneration in FOP was recently investigated by applying iPSC technology [126]. The authors aimed to recapitulate observations in tissues from cadaveric specimens using non-isogenic control and FOP iPSCs differentiated into myogenic progenitors combining small molecules (i.e., CHIR-99021, LDN-193189) and growth factors (i.e., bFGF). The mRNA transcriptome of primary human satellite cells (Hu-MuSCs) and iPSC-derived muscle stem/progenitor cells (iMPCs), from control and FOP donors, unveiled that iMPCs do express a more progenitor-like phenotype as compared to Hu-MuSCs (Table 1). While the FOP Hu-MuSC engraftment and regeneration capacity was compromised, this was not the case for the control and FOP iMPCs, which performed similarly. In vitro, both FOP iMPCs and Hu-MuSCs expressed high levels of the ALK2 target genes *ID1* and *ID3*, as well as chondrogenic and ECM genes. However, neither iMPC nor Hu-MuSC implants formed HO in vivo, suggesting that other cell types are the ultimate bone mediators in FOP. 

Since inflammation triggers bone formation in FOP, recent advances in iPSC differentiation protocols have enabled the study of the impact of ALK2^R206H^ on monocyte activation and cytokine secretion. Matsuo et al. utilized two distinct macrophage differentiation protocols from iPSCs, both in 2D and 3D formats [127]. Although a direct comparison between control and FOP cells was not possible in the new 3D embryoid body-based protocol, the 2D differentiation method, using a commercially available kit, yielded a heterogeneous population of M1/M2 macrophages. These cells could be efficiently polarized into M1 or M2 states using specific stimuli (Table 1). Notable differences were observed in FOP iPSC-derived M1 macrophages, which exhibited higher expression levels of IL6, IL1a, RANTES, and Activin-A, among others. Maekawa et al. pursued a different approach by differentiating isogenic iPSC lines into the monocytic lineage and subsequently immortalizing the cells [128,129]. FOP immortalized monocytes (FOP-ML) displayed elevated levels of phosphorylated SMAD5, both at basal levels and upon stimulation with recombinant Activin-A, unlike the control cells. However, neither study investigated the potential differentiation of monocytes into osteoclasts, which play a crucial role in bone remodelling.

The implementation of iPSC technology into FOP research has led to major advances in drug development for this so far uncurable disease. Experiments performed in isogenic control and FOP-derived mesenchymal stromal cells (MSC) unveiled what seems to be the underlying mechanism driving FOP. Using a BMP transcriptional reporter assay, Hino et al. screened for most of the TGF-β superfamily ligands to discover that recombinant Activin-A induced BMP signalling and downstream chondrogenic differentiation exclusively in FOP cells [130]. Almost simultaneously, these findings were reported in humanized murine embryonic stem cells from a newly developed FOP model, where pharmacological sequestration of circulating Activin effectively prevented extraskeletal bone formation [123]. Currently, Garetosmab (a humanized anti-Activin antibody) is being clinically evaluated for its potential to prevent HO in individuals with FOP (clinical trial NCT05394116). In a different study, FOP iPSCs, incorporating a novel reporter construct based on the *COL2A1* promoter and the *aggrecan* enhancer driving luciferase expression, were combined with drug-repurposing chemical libraries. Screening nearly 7000 compounds identified the potential therapeutic value of targeting mTOR signalling in FOP, particularly with rapamycin [131,132]. These findings have led to the initiation of the first clinical trial based on iPSC-derived cells (Japan, UMIN000028429) investigating the repositioning of rapamycin for FOP. A more recent clinical trial aims to evaluate the application of Saracatinib in FOP (clinical trial NCT04307953). Identified through the same screening unveiling rapamycin [132], Saracatinib has been repurposed as a selective ALK2 inhibitor, preventing Activin neo-function in iPSC-ECs [133]. This and other manuscripts have found how aberrant Activin signalling in FOP patients (Figure 3) is recapitulated not only in FAPs, but also in other cell types such as dermal fibroblasts, urine epithelial cells, blood-derived monocytic and endothelial cells, and periodontal ligament fibroblasts [112,116,134,135,136,137,138]. Using different primary or iPSC-derived cell types may facilitate the investigation in organ-specific Activin mechanisms, which is useful for characterizing novel druggable targets and/or evaluating the potential toxicity of drugs under investigation. 

### 2.2. Hereditary Pulmonary Arterial Hypertension and hiPSCs

Pulmonary arterial hypertension (PAH) is a rare disease characterized by the narrowing of arterial blood vessels that carry deoxygenated blood from the right side of the heart to the lungs. PAH has been linked with gene mutations in the BMP signal transduction pathway (e.g., *ACVRL1*(ALK1), *BMPR1B* (ALK6), *ENG* (endoglin), and *SMAD9*)*,* but also gene mutations in genes outside of the BMP pathway have been found among those patients with and without familial forms of PAH (i.e., *CAV1*, *KCNK3*, *EIF2AK4*). However, *BMPR2* mutations are predominant, accounting for 70–80% of all cases of heritable PAH (HPAH, formerly familial PAH) [139,140,141,142,143,144] (Figure 3). Different HPAH-causing variants, mapping in different parts of the *BMPR2* gene (including its promoter region), have been described, resulting in a loss of BMPR2 expression, abnormal trafficking [145], mislocalization [146], and/or a haploinsufficient phenotype [147,148]. Moreover, some mutations give rise to dominant negative BMPR2 proteins due to mechanisms escaping nonsense mediated decay [149,150,151]. BMPR2 expression is ubiquitous, but high affinity complexes predominantly with ALK1 are formed mainly in the vasculature. ALK1 cooperates with the co-receptor endoglin, strongly expressed in the endothelium [152,153], to further enhance ligand binding affinity and receptor downstream signalling [154,155]. Moreover, endoglin is required to integrate extracellular mechanical forces, such as shear stress exerted by blood flow, into the BMP signalling pathways [65,66,156,157]. In ECs, BMP-9/-10 induce SMAD1/5/9 signalling and expression of vascular cell specific target genes [158], through ALK1-ENG-BMPR2, in order to maintain EC quiescence and expression of endothelial specific genes. It is therefore not surprising that next to *BMPR2* mutations, *SMAD1*, *SMAD9, ACVRL1* (ALK1), and *ENG*, mutations can also drive HPAH (reviewed in [140]). The latter two (*ACVRL1* and *ENG* mutations) are the main heritable risk factors for HHT, which will be covered in the next paragraph. For simplicity reasons, we will focus on *BMPR2* mutations and HPAH (Figure 3). 

BMP-9 and BMP-10 are blood circulating factors. Gene regulation by BMP-9/-10 signalling provides the endothelium with survival and quiescence signals [159,160], as well as inhibiting vascular permeability [161]. This is particularly important under challenging environmental conditions of the vasculature such as hyperglycaemia [161], hypoxia [162], or inflammation [163], which are all known to induce hyperpermeability of the vasculature. BMPRs interact with a number of vascular-specific transmembrane proteins in arterial and venous blood vessels and lymphatics, including VEGFR2 and neuropilins (NRP1/2), reviewed in [164]. Furthermore, BMP-9/-10-BMPR2-ALK1 signalling protects ECs from undergoing EndMT, which involves EC activation and extensive transcriptional reprogramming, and leads to a shift of mature ECs towards a mesenchymal phenotype [119,165,166] (Figure 3). ECs undergoing EndMT can adopt a myofibroblast-like state where cells tend to deposit aberrant fibroblast-like ECM, which changes their own and the environmental mechanical features [167,168,169,170]. We will further expand on these aspects in Section 3.1. In smooth muscle cells (SMCs), BMPR2 signalling protects their hyperproliferative growth responses towards TGF-β [171,172,173] and maintains their contractile state [58]. The major cellular consequences of HPAH are found in ECs and SMCs of the pulmonary artery vasculature, where disruption of BMP-9/-10-BMPR2-ALK1-endoglin signalling causes, as a consequence of the beforementioned cellular functions, EC apoptosis, EndMT, and SMC hyperproliferation. This results in the narrowing of the pulmonary arterial lumen of the arterial wall, through a process called neointima formation, and the increase in vessel wall thickness, through a process called medial hypertrophy (Figure 3). The resulting increase in lung blood pressure not only has negative effects for the perfusion of the lung itself but also for the right heart chamber coping with the increased arterial resistance. Unfortunately, most rodent models of HPAH do not precisely recapitulate the disease pathology; these models display less substantial pulmonary vascular remodelling in both proximal arteries and distal microvasculature, which significantly hampers drug discovery efforts [174,175,176]. Presently, there are number of drugs that have been approved by the Food and Drug Administration (FDA) for the treatment of HPAH and its idiopathic form (IPAH), whilst most address only the clinical symptoms limited to vasodilator treatment. No FDA approved PAH drug is currently able to reverse the described vascular remodelling. Recent approaches discovered that the use of the immunosuppressive drug FK506 is able to inhibit the interaction between the BMP type I receptor kinase domain and the endogenous intracellular BMP signalling inhibitor FKBP12 [177]. More recent advancements have focused on targeting HPAH by addressing the elevated levels of Activin-A using the Activin-ligand trap Sotatercept/ActRIIa-Fc [63,178,179]. This progress in drug development was partly due to the use of iPSC technology.
Figure 3**Rare BMP diseases and iPSCs**. Fibrodysplasia ossificans progressiva (FOP) is a musculoskeletal disease caused by heterozygous gain-of-function mutations in *ACVR1*/ALK2, resulting in heightened activation of downstream ALK2 signalling in response to Activin-A. Increased ALK2 signalling leads to the formation of extraskeletal bone plaques. On the right is a photograph of the skeleton of a man with FOP, from the collections of the Anatomical Museum of the Leiden University Medical Center (LUMC). This photo has been reproduced with permission from LUMC (copyright belongs to LUMC). Hereditary pulmonary arterial hypertension (HPAH) is a cardiovascular disease characterized by progressive narrowing of the pulmonary arteries, resulting in elevated pulmonary arterial pressure (mPAP > 20 mm Hg), followed by right ventricular dilatation and hypertrophy. Loss-of-function mutations in BMPR2 exacerbate disease severity and penetrance by disrupting the overall balance of TGF-β signalling in endothelial and smooth muscle cells. The right panel illustrates a cross-section of a healthy pulmonary artery (top) compared with a diseased occluded pulmonary artery (bottom). The formation of a pronounced neointima by aberrant cell growth into the lumen is shown. The PAH patient pulmonary arterial cross-section is part of a published dataset by C.H., recreated based on details mentioned in Hiepen C. et al. [168]. Hereditary haemorrhagic telangiectasia (HHT) is associated with loss-of-function gene mutations in *ACVRL1* (ALK1) and *ENG* (endoglin), among others, which impair BMP signalling. It is characterized by varying degrees of lesions due to the instability of the microvascular networks. The right panel displays a typical microvascular subdermal lesion seen in HHT. The picture was contributed with friendly permission by Prof. Dr. Urban Geisthoff (Philipps University of Marburg; Germany).
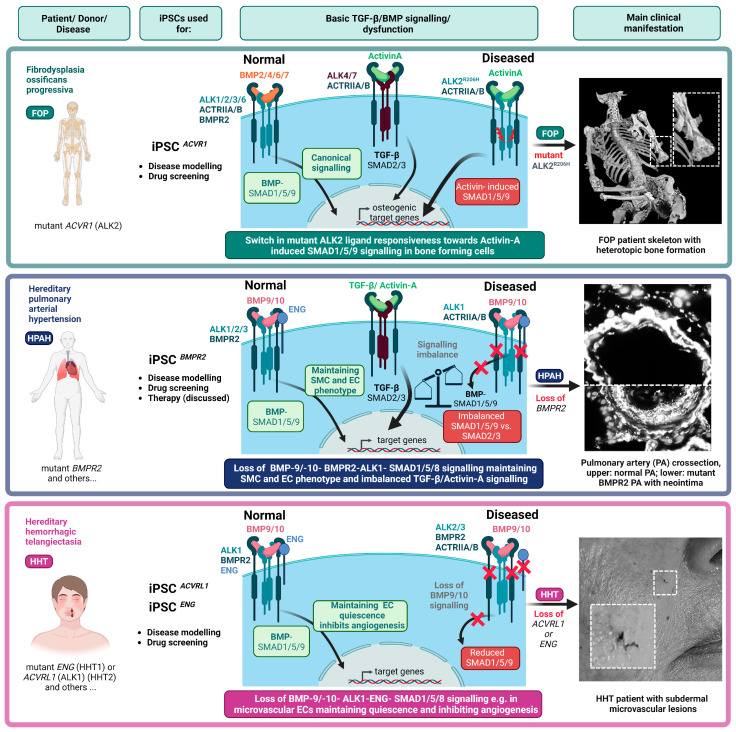


For HAPH disease modelling and drug development, human ECs that are typically isolated from umbilical-cord vein (HUVECs) may not represent the appropriate test system, since they are juvenile, form rather large veins, and have not experienced the same oxidative and mechanical environment as pulmonary arteries. Moreover, their access is limited, the cost of commercial HUVECs is high, and batch-to-batch reproducibility is often too low. iPSC-derived cell types can serve as an unlimited source for surrogates for different EC types, including veinous ECs, arterial ECs, (brain) microvascular ECs, and lymphatic ECs, in both functional and drug discovery studies [180,181,182]. The first HPAH-patient-derived iPSCs were generated in 2012 at the University of Cambridge by Geti, Vallier, Morrell, and co-workers [183] (Table 1). They used late-outgrowth endothelial progenitor cells (L-EPCs) from peripheral blood mononuclear cells (PBMNC) as a somatic source for iPSC reprogramming. EPCs possess several favourable characteristics as a somatic cell source for the generation of iPSCs over, e.g., dermal fibroblasts. Although skin fibroblasts are the most common cell type used for generating iPSCs, their isolation requires a surgical procedure, which is also undesirable in children, patients with skin disorders, and patients with abnormal coagulation or wound healing. In addition, fibroblasts are reprogrammed with relatively low efficacy. EPCs are also readily generated from frozen PBMNC preparations, making storage and transportation of somatic source material straightforward. As a proof-of-principle, they generated EPC-iPSCs from both healthy individuals and patients with heritable *BMPR2* mutation and idiopathic PAH. Compared with the traditional approach using fibroblasts, EPCs displayed high reprogramming kinetics and efficiency, taking only 10 days to emerge in culture and form iPSC colonies. This first generation of PAH-derived iPSC was followed by the first directed differentiation approach, conducted at Vanderbilt University by West et al. in 2014 [174], in which they applied mesenchymal and EC differentiation protocols combined with a transcriptomic profiling of iPSCs and differentiated cells, respectively (Table 1). The *BMPR2* mutation resulted in changes in genes that were consistently altered by the mutation, irrespective of the cell differentiation state. These included genes, e.g., associated with Wnt signalling. However, the authors also identified genes that were exclusively differentially expressed in mutated cells only upon their differentiation into specific lineages, such as secreted frizzled-related protein 2 [174]. This study was followed by work from the Rabinovitch lab in Stanford in 2017, aiming at identifying endogenous mechanisms that may revert the impact of the *BMPR2* mutation [184,185]. This is particularly important because only 20% of the *BMPR2* HPAH mutation carriers develop clinical symptoms, and it is not known why. For this, Gu et al. compared the transcriptomes of iPSC-derived ECs from unaffected BMPR2 mutation carriers with those from HPAH patients with clinical symptoms. The same study also used CRISPR/Cas9 for *BMPR2* gene correction in patient-derived cells for the first time, generating isogenic controls [184]. Upon differentiation, the corrected HPAH iPSC-EC lines showed levels of cell adhesion molecules and survival rates in response to either serum withdrawal or after hypoxia and reoxygenation, which were similar to control cells (Table 1). In accordance with many other studies, they could show, e.g., that, in iPSC-derived ECs from HPAH patients, reduced BMPR2 expression led to impaired canonical BMPR2 signalling (pSMAD1/5-ID1) in response to BMP treatment (Figure 3). The RNA-sequencing dataset from this study is valuable and continues to be used in the identification of new *BMPR2* mutation-associated mechanisms [186]. Sa et al. compared pulmonary arterial (PA) ECs and iPSC-derived EC from the same HPAH patients, since several studies suggest that PA cell-specific features (such as differential chromatin methylation patterns in the endothelial nitric oxide synthase genes) contribute to the development of PAH [185]. They also generated PAEC and iPSC-derived ECs from patients suffering from idiopathic PAH, in order to find communalities between IPAH and HPAH and to develop personalized medicine approaches. One of their major findings was that IPAH, HPAH-PAECs, and iPSC-derived ECs from the same donors exhibit similar reduced gene expression of collagen type IV (*COL4A2*), a major ECM protein of the vascular wall basal lamina. Subsequently, Gu et al. conducted an extensive study utilizing patient-derived iPSCs, followed by endothelial cell (EC) differentiation, to identify potential new drug candidates. This was accomplished through high-throughput drug screening combined with in silico analysis of existing transcriptomic datasets. They identified a promising lead compound named AG1296 (Tyrphostin) [187]. Tyrphostin is a tyrosine kinase inhibitor targeting platelet-derived growth factor (PDGF), c-Kit, and FGF receptor signalling, and it has potent effects on cells nanomechanical properties and cytoskeletal organization [188,189,190]. Kiskin et al. has recently generated iPSC-SMCs that recapitulated some of the important functional responses of adult-derived distal pulmonary arterial smooth muscle cells (PASMCs), as well as iPSC-ECs with enhanced expression of arterial specific markers [191] (Table 1). They obtained control and *BMPR2* mutant isogenic lines using CRISPR/Cas9. This revealed, for the first time, that iPSC-derived PASMCs from HPAH patients resemble the hyperproliferative phenotype observed in primary cells. They also used those iPSC-derived PASMCs and arterial ECs to investigate the role of BMP-9 as a PAH reversing agent [162,192]. Using the same model, the role of TNFα as an important secondary driver of the PAH disease mechanism was analysed [63]. In a recent study released as preprint, HPAH patient derived iPSCs were also differentiated into vascular SMCs [193]. The study suggests that progesterone promotes the proliferation and migration of HPAH iPSCs-derived VSMCs, supporting a model that describes why penetrance for *BMPR2* mutation carriers is observed with a female-sex bias [194,195,196]. 

Thus far, the use of hiPSCs has significantly contributed to the discovery of BMPR2 insufficiency being the major heritable risk factor for HPAH and also to the identification of underlying molecular pathomechanisms. Moreover, iPSCs from HPAH patients allowed for the identification of promising new target genes and drug candidates. Using iPSCs for stem cell therapy in PAH has also been discussed [197] (Figure 4). The first experiments in animals have suggested a therapeutic effect of injecting either iPSCs, iPSC-derived conditioned medium, or iPSC-derived exosomes in animal models of PAH [198,199]. Although further experimental validation in several preclinical models is desired, these results underline the benefits of paracrine effects in treatment of PAH.

### 2.3. Hereditary Haemorrhagic Telangiectasia and hiPSCs

Hereditary haemorrhagic telangiectasia (HHT) is an autosomal dominant disorder mainly affecting the microvasculature, including capillaries, venules, and arterioles. Three major genes are causally related to HHT: the *ENG* gene encoding the co-receptor endoglin (ENG/HHT1) [200]; the *ACVRL1* gene encoding ALK1 [201,202,203] causing HHT2; and the *MADH4* gene (SMAD4) (HHT3), a critical intracellular mediator in BMP and TGF-β signalling pathways [202,204] (Figure 3). *GDF2* gene mutations (encoding for BMP-9) causing HHT [205] are extremely rare. For HHT2, about 350 different *ACVRL1* mutations are known, including deletions, splice site mutations, and missense mutations of *ACVRL1*, the latter building the largest group. HHT patients exhibit characteristic severe epistaxis (nose bleeding) and haemorrhage of the subdermal microvasculature, which is called “telangiectasia” [206] (Figure 3). Other capillary beds are affected by haemorrhages and dilations, including mucosa-associated blood vessels in the nose, liver, lung, gut, and brain. Haemorrhages are due to the loss of the intervening capillary bed and the formation of dysfunctional arteriovenous malformations (also known as “shunts”) associated with a lymphocytic perivascular cell infiltrate. Those rather less harmful appearing vascular malformations can become very severe, affecting the blood flow in the brain and the heart. HHT patients often suffer from severe iron deficiency and anaemia [207]. In *Acvrl1* defective HHT rodent models, it was shown that local injury triggers an abnormal re-vascularization with dilated and tortuous vessels. In mice, this might be due to excessive sprouting angiogenesis, in line with previous reports [208]. Cell–cell communication seems deficient in HHT ECs, compromising the pericyte coverage of the microvascular endothelium, which is also required for vessel stabilization and maturation [209,210]. While the role of sprouting angiogenesis to HHT development is currently under investigation, one possible explanation for increased angiogenic responses of *ACVRL1* mutant ECs may be that their pericyte coverage is reduced, lacking a barrier that protects ECs from direct exposure to pro-angiogenic stimuli that induce angiogenic sprouting from pre-existing blood vessels, e.g., as observed during vascular development [211]. To date, a precise understanding of how dysfunctional BMP-SMAD1/5/9 signalling is involved in HHT is missing. Current models suggest that BMP-9/-10-ALK1-endoglin-SMAD target genes do inhibit angiogenesis [212,213] required to protect against HHT development (Figure 3). However, non-canonical BMP signalling may also contribute to the HHT pathomechanism. Moreover, there might be a link with the flow perception of ECs through endothelial primary cilia, which are flow-sensitive organelles composed of microtubules. Primary cilia act as signalling hubs that connect the mechanical forces of hemodynamic flow to canonical BMP/TGF-β family signalling [66,214,215,216]. The loss or dysfunction of endothelial primary cilia could potentially contribute to the development of arteriovenous malformations (AVMs) by affecting ALK1-SMAD signalling [217]. This only highlights the role of the local pro-angiogenic and mechanical microenvironment in the formation of arteriovenous malformations. Currently, the most effective treatments for HHT involve repurposed anti-angiogenic drugs that are already utilized in cancer therapy. These include anti-VEGF antibodies like Bevacizumab, which is a recombinant humanized monoclonal antibody that hinders blood vessel formation by inhibiting VEGF [218,219]. Additionally, several tyrosine kinase inhibitors are also being investigated as potential treatments [220].

The first approach to generate patient derived HHT iPSCs was published by Freund et al. at the Leiden University Medical Centre, in the Netherlands [221]. These efforts were followed by generation and genetic repair of two iPSC clones from a patient bearing a heterozygous *ACVRL1* gene leading to HHT2. Correction was performed with CRISPR/Cas9 gene editing, creating control iPSCs from the same patient [222]. Zhou et al. generated isogenic control and HHT1 iPSCs from a patient with an *ENG* mutation. The iPSCs derived from this patient, carrying a novel missense mutation, displayed reduced potential for differentiation into iECs, which performed poorly in tube formation assays. Furthermore, the C30R *ENG* mutation resulted in the retention of the mutated endoglin protein in the endoplasmic reticulum, possibly due to protein misfolding. Functionally, BMP-9 stimulation resulted in reduced phospho-SMAD1/5-ID1 signalling in comparison to corrected cells, confirming in human iPSCs, for the first time, that mutant endoglin affects BMP-9 downstream signalling. Moreover, their HHT iECs displayed a disorganized cytoskeleton which, after CRISPR/Cas9 mediated correction, turned into a highly organized cytoskeleton [223]. This recapitulated the previous findings by others using primary blood outgrowth ECs isolated from HHT patients [224]. Very recently, Xiang-Tischhauser et al. established HHT2 iPSCs through gene editing an *ACVRL1* mutation in a parental iPSC-clone, causing a frameshift mutation that led to reduced ALK1 expression. In those mutant ALK1 iPSCs, the expression levels of *ENG*, SMAD2/3, and TGF-βRII were not notably different between parental and mutant iPSC lines; however, they found upregulation of VEGFA and noggin expression. Their gene-edited iPSCs were also differentiated towards ECs and to form embryoid bodies (EBs) [225]. An inspiring work by Orlova et al. underlines the importance of iPSC-derived cell culture under advanced environmental conditions to better recapitulate the human disease complexity. When using 2D cell culture models, control and HHT1-isogenic iECs and pericytes were able to form indistinguishable vascular networks. However, when grown in 3D microfluidic vessel-on-chip devices with fibrin hydrogel and under flow, lumenized vessels formed, in which defective vascular organization was evident [226]. The authors found a leaky 3D vascular network in this iPSC-derived vessel-on-chip model reminiscent of what was reported from HHT patients. The interaction between iECs and primary human brain pericytes that were added to the microfluidic system was disturbed. These results highlight that iPSC culture under organ or tissue-specific conditions can be required for some BMP-related rare disease models in order to mimic cellular responses to a particular genetic defect recapitulating major aspects of the human phenotype. 

Interestingly, a few HHT individuals carrying *ACVRL1* mutations are predisposed to the development of PAH [227]. Currently, in those *ACVR1L* mutant affected HHT2 patients, PAH is seen as an extremely rare complication [228]. Similarly, *GDF2* mutations cause a vascular-anomaly syndrome somehow overlapping with HHT [205], and some *BMPR2* variants increase the risk of developing pulmonary AVMs reminiscent of an HHT-like condition [229]. Vice versa, it is not surprising that *ACVR1L* mutations can also cause PAH-like and HHT symptoms [228,230] in the same patient. In that respect, more research is required to understand why the same mutation can cause different vascular abnormalities in different vascular beds in some patients but are also very rarely combined in the same individuum.
cells-12-02200-t001_Table 1Table 1Overview on published BMP-related rare disease iPSC models.Rare DiseaseMutation Somatic Source Cell Age and Sex of Donor (s)Re-Programming MethodDifferentiation ProtocolReferenceFOPACVR1c.617G>A (p.R206H)c.1067G>A(p.G356D)Dermal fibroblasts18-M,59-F,22-M,66-FSendaiSpontaneous differentiation into mesodermal and endodermal lineages[114]FOPACVR1c.617G>A (p.R206H)Dermal fibroblasts16-F,16-F,∅-M∅-M12-M,12-M,50-MRetroviral,Episomal Osteoblast-like mineralization,chondrogenicEC: [121]2D and 3D chondrogenic: [130]Presomitic mesoderm and somites [125]Muscle stem/progenitor cells: [126][115]FOPACVR1c.617G>A (p.R206H)Kidney cells isolated from urine∅-M∅-MEpisomalEC[116]FOPACVR1c.617G>A (p.R206H)Urinary cells∅-M∅-FSendaiEC: [133][231]FOPACVR1c.617G>A (p.R206H)Periodontal ligament fibroblasts 23 yearsSendaiSpontaneous differentiation into EBs[136]HPAHBMPR2c.27G>A (p.W9X)c.1039T>C (p.C347R)Late-outgrowth endothelial progenitor cells from vein blood∅-∅∅-∅RetroviralDifferentiation into 3 germ layers and teratoma[183]HPAHBMPR2c.354T>G (p.C118W)c.2504del (p.T835fs)c.350G>A p.C117>YDermal fibroblasts
Lentiviral/Sendai EC[184]HPAHBMPR2c.1471C>T (p.Arg491Trp)Dermal fibroblasts37-M, 33-F
EC[187]HPAHBMPR2 c.1471C>T (p.Arg491Trp)Dermal fibroblasts37-M, 33-FSendai EC[185]HPAHBMPR2W9X+/−C2 ΔExon1Parental wt iPSC line genome edited via CRISPR/Cas9∅-∅
SMCEC[191]HPAHBMPR2c.1598A>G (p.His533Arg)c.248-1–418+1del171(p.Cys84_Ser140del)c.1471C>T (p.Arg491Trp)CD34^+^ blood cells38-M, 17-F, 5-F Sendai In vitro spontaneous differentiation into 3 germ layers[232]HHT2ACVRL1(ALK1)Dermal fibroblasts∅-∅RetroviralCardiomyocytes by END-2 co-culture[221]HHT2ACVRL1 c.1120del18Dermal fibroblasts42-MEpisomalIn vitro spontaneous differentiation and directed trilineage differentiation[222]HHT2ACVRL1c.772 + 3_772 + 4dupPBMCs61-FSendaiIn vitro directed trilineage differentiation[233]HHT2ACVRL1heterozygous frameshift deletion of 17 base pairs in exon 8Fibroblast-derived parental iPSC line ATCC; CRL-2097 genome edited via CRISPR/Cas9Neonate (<1 month)-MSendaiEmbryoid bodies and ECs [225]HHT-1ENGc.88T>C (p.C30R)PBMNCs62-FEpisomalEC[223]HHT-1ENGc.1678C>T p.(G560∗)PBMCs∅-MEpisomal ECco-culture with primary human brain vascular pericytes [226]An overview of published BMP rare disease hiPSC models is given: Rare disease abbreviation (column 1), gene mutation and consequences for protein (column 2), somatic source-cells used for reprogramming (column 3), age and sex (column 4), reprogramming method (column 5), established differentiation protocols (column 6), references (column 6); ∅ = not disclosed, ∗ = translation termination, M = male, F = female, EBs = embryoid bodies.

## 3. Culturing iPSCs under (Patho)Physiologically Relevant Microenvironmental Conditions

Traditionally, iPSC studies use 2D monolayer cultures combined with specific differentiation conditions, consisting in basal media supplemented with defined concentrations of small molecule inhibitors and differentiation factors. A 2D culture is conducted on an iPSC compliant culture surface, such as basement membrane coating. However, to accurately mimic BMP-related rare disease mechanisms in vitro, 3D culture conditions and the combination of different cell types in a biomechanically challenging environment may be required, as demonstrated by Orlova et al. [226] (Figure 4). Achieving this necessitates expertise in (stem) cell biology, TGF-β and/or BMP signalling, cell-material interactions, mechanobiology, and the biochemical context of diseased cells in affected tissues. To replace animal disease models with human iPSC technologies, complex in vitro culture conditions using devices like microfluidic chips and bioreactors are needed to address these aspects. These systems should resemble the complex tissue architecture, circulation, and systemic effects. Current efforts aim to develop fully integrated, controlled, and sensor-equipped systems capable of simultaneously studying multiple cell types [234,235]. Microfluidic organ- and body-chips represent a promising model for iPSC-derived cell culture under complex 3D conditions. Co-culture of iPSC-derived cells in such devices allows the study of the interplay between different cell types from a common genetic background. iPSC-derived cell culture can be combined with tissue engineering and bioprinting of three-dimensional cell constructs, organoids, and grafts [236,237] (Figure 4). Based on person-centred biomedical data of the somatic cell donor (e.g., age, sex, ethnics, co-morbidities, medication, medical history), the goal for patient-specific disease modelling, as well as personalized drug screening and personalized medicine, may be to combine microfluidic organ-on-chip technology with BMP rare disease patient iPSC-derived cells in a 3D matrix (organoids), following the concept of “miniature twinning” [238,239]. Organ-on-chip, 3D organoid-on-chip, or multi-organ-chip systems currently attract a lot of attention [240,241] and have been successfully applied in recent years to create complex laboratory model systems for, e.g., systemic COVID-19 research [240], blood–brain barrier function [242], and bone regeneration [243,244]. The integration of biosensors into those high-tech systems and the concomitant application of defined cell mechanics make them an even more versatile tool with which to study disease mechanisms and accelerate drug development (Figure 4).

### 3.1. Microenvironmental Aspects Relevant for Mimicking BMP Rare Diseases with iPSCs

The Knudson hypothesis, also known as the “two-hit” hypothesis, states that most tumour suppressor genes require both alleles to be inactivated [245]. Considering additional genetic variations that act in addition to the primary genetic event, PAH and HHT can be considered “two or second-hit diseases” as well [246,247,248,249,250]. Both come with low penetrance, and not every *BMPR2* and *ALK1*/*ENG* mutant carrier develops clinical symptoms. Multifactorial inputs seem required for PAH and HHT onset and progression, beyond genetic disruptions. For BMP rare diseases like PAH and HHT, the concept of the “genetic” second hit hypothesis could be extended towards microenvironmental cues, eventually cell/bio mechanics, hypoxia, and inflammation, that add to disease onset and progression as a “non-genetic second-hit” [66,170,216,251,252]. Any iPSC-derived in vitro model, aiming at resembling the human PAH and HHT pathomechanism as closely as possible, may consider that.

In FOP, the combination of inflammatory tissue trauma and the genetic *ACVR1* mutation is necessary to drive extraskeletal ossification, strongly suggesting such non-genetic second-hit dependency as well. The expression of *INHBA*, encoding for the Activin-A homodimer, is induced by inflammatory signalling pathways [253], but systemic Activin levels are not increased in FOP individuals [254], suggesting a strong role for the local cellular FOP microenvironment versus systemic factors. Interestingly, recent single cell RNA sequencing data obtained from HO lesions in FOP mice indicates that Activin mRNA expression is induced locally [255]. Congruently, a recent report has shown how dermal fibroblasts from FOP donors express high levels of Activin-A compared to controls [138], and, as aforementioned, iPSC-derived inflammatory macrophages showed a significantly higher production of Activin-A [127]. This might suggest that specific local tissues or organ environments may modulate functional cell responses in FOP. Intriguingly, this has been previously discussed not only in FOP, but also in PAH and HHT [256,257,258,259]. An intricate feedforward regulatory dependency exists with regard to the local cellular ECM-based biomechanical microenvironment and how this may be influenced by mutations in the BMP pathway. Haupt et al. showed that progenitor cells from FOP-like mutant mice misinterpret the tissue microenvironment through altered sensitivity to mechanical stimuli, which lowered the threshold for commitment into chondro/osteogenic lineages [260]. Evidence suggests that *ALK2^R206H^* mutant cells contribute to an active stiffening of the surrounding ECM, e.g., by increased collagen production [121,126]. Members of the TGF-β ligand family were shown to induce osteogenic differentiation of mesenchymal progenitors more potently on stiff substrates [261]. Altered BMP-SMAD signalling influences the expression of ECM genes in various cell types, leading to changes in the mechanical microenvironment. Increased ECM deposition, e.g., affects the local stiffness and elasticity, determined by ECM composition and degree of crosslinking. The cells elaborate feedback responses to these changes through outside-in signalling, primarily mediated via integrins, which, themselves, interact with—and crosstalk to—BMPRs [262]. Integrins enable cells to transduce feedback from alterations in ECM stiffness into their signalling pathways, a phenomenon generally observed in TGF-β family signalling (reviewed in [263]). Interestingly, ECM changes, actin-cytoskeletal rearrangements, and mechanical features such as changes in cell stiffness, traction forces, and actomyosin contractility show commonalities between all three mentioned BMP rare diseases: FOP, PAH, and HHT [168,260,264,265,266]. Particularly, pathological defects on the cell cytoskeletal organization, cell morphology, migration behaviour, and adhesion molecule expression influence cell–cell and cell–substratum traction forces [267,268]. In PAH, ECM deposition and cytoskeletal defects are likely to contribute to pulmonary artery stiffening, constriction, and autocrine TGF-β signalling through increased retrieval of mature TGF-β from ECM-bound latency complexes [168]. In HHT, blood flow and changes in cellular biomechanics may affect cell junction tightness and pericyte coverage. Analogies can be drawn between HHT and the role of biomechanics in the field of cerebral cavernous malformations, where the intricate relationship between vascular malformations and dysplasias with EC mechanotransduction is better understood [269].

The important role of fluid shear stress and the mechanotransduction of vascular cells for PAH and HHT development was shown [66,270], and mechanical stimulation of different cell types can influence BMP signalling outcomes [52,65,271,272] (Figure 4). Endoglin is required to implement extracellular mechanical forces, such as shear stress exerted by blood flow, into the BMP signalling pathways [66,156,157]. The endoglin role in flow-perception during the HHT pathomechanism is currently investigated by several laboratories. To address shear stress by (micro)fluidic systems in both PAH and HHT, channel geometries and diameter, flow rates, viscosity of the medium, and the particular type of wall shear stress need to be considered. The shear stress values in arterial vessel types can be very different (5–50 dyne/cm^2^), and wall shear stress can follow different flow regimes [273]. During turbulent flow, regions of flow recirculation or flow separation can occur, which are more challenging to reproduce in a microfluidic device. Understanding the exact contribution of fluid shear stress and its integration into BMP signalling may help explaining why a single mutation (e.g., *ALK1* or *ENG*) can lead to two distinct rare disease phenotypes (PAH and HHT) in different vascular beds (pulmonary arteries vs. microvasculature). The flow characteristics in the microvasculature differ significantly from those in arteries and veins impacting oxygen levels [274]. Additionally, arterial, venous, and microvascular ECs exhibit distinct responses to blood flow [275]. Therefore, in order to replicate PAH and HHT using microenvironmental mechanical shear stress cues, it is essential to consider the appropriate perfusion systems that allow for different flow rates and regimes, oxygen conditions, and the culture of specific subtypes of EC, eventually even in the presence of SMC or pericyte co-cultures (Figure 4). Several protocols have already been developed to differentiate iPSCs into arterial-like, venous-like, and microvascular-like ECs, and these have been combined with microfluidic platforms that mimic arterial, venous, and microcirculatory perfusion [180,276,277]. In summary, although iPSC technology has made progress in the study of BMP-related rare diseases, there is still a long way to go in establishing complex in vitro platforms that accurately mimic the biochemical and biomechanical cellular processes involved in the onset and progression of these pathologies.

## 4. Future Perspectives

The future holds iPSC-derived BMP rare disease models that can be cultured under complex microenvironmental conditions aiming at resembling closely the pathological in vivo conditions. This involves incorporating inflammatory cues, disease-specific mechanical stimulation, and a permissive microenvironment for optimal disease mimicry. In terms of BMP signalling interrogation, accurately recapitulating disease (micro)environments may not only directly impact canonical BMP-SMAD pathways, but also the crosstalk with other branches of the TGF-β superfamily and/or other non-canonical signalling cascades. As such, “non canonical” and/or “non-SMAD” signalling must be better defined and further explored [278]. While a number of perturbed signalling mechanisms related to FOP, PAH, and HHT were already discovered, further insights on diversification and cell-type specific BMP signalling modes may increase by use of iPSC technology and its combination with CRISPR/Cas9, e.g., by the design of smart biosensors and a higher resolution of the applied analytics. As such, the discovery of “lateral SMAD” signalling, i.e., the activation of BMP-SMADS by TGF-β or Activins via altered receptor complex stoichiometry, is of current interest for the mentioned pathologies [279]. This only emphasizes the need for comprehensive studies aiming to integrate input from different layers, to dissect complex regulatory networks, and to identify novel disease biomarkers or druggable targets. To overcome this, almost all omics methods are rapidly developing towards increasing depth, robustness, automatization, and affordability, while requiring smaller sample yields. These characteristics will ultimately facilitate the implementation of human iPSC-based miniaturized disease models as routine preclinical platforms in biomedical research. The FDA no longer requires animal testing for new drugs entering human clinical trials, which is also addressing ethical concerns [280]. While a number of transgenic animal models exist for studying BMP rare diseases, they may not fully replicate human disease phenotypes. However, patient-derived iPSC disease models also present challenges, i.e., obtaining rare-disease patient material, scalability, stability, automation, and material selection [281]. Even though the implementation of CRISPR/Cas9 to generate control and disease lines with an identical genetic background has substantially improved the field, other aspects of culturing and differentiating iPSCs are still a source of high variability. For example, substrate stiffness impacts stem and progenitor cell differentiation [282]. When selecting substrates for iPSC maintenance, differentiation, or organoid formation, considering their stiffness and the effect on differentiation is important [283,284]. Various hydrogel systems using natural or synthetic materials have been developed to mimic the stem cell niche in vitro, providing biochemical and biophysical signals. The choice of the right ECM for culturing iPSCs, whether in 2D or 3D organoids, can impact ECM secretion and formation by the cells. Considering the potential involvement of ECM in disease mechanisms such as FOP and PAH, caution is necessary when selecting the right ECM for BMP rare disease modelling. ECM extracts have batch-dependent variability, and their polymerization is influenced by multiple factors. The sequestration of recombinant BMPs by natural ECM proteins used as iPSC substrates can reduce the effective concentrations required for BMP stimulation of iPSCs towards mesenchymal differentiation [285]. Instead, current research focuses on designer matrices for iPSC culture [286,287], which offer better reproducibility and adaptability to, e.g., bio(ink)-based printing technologies (Figure 4). Importantly, every new technology development that claims to represent an appropriate animal-free alternative system for disease modelling needs to demonstrate that the cellular model is functionally and phenotypically representative of the native cell/tissue. For this, more comparison and referencing towards, e.g., patient-derived tissue samples is required. Finally, international harmonization and implementation of quality control standards concerning iPSC research with regard to iPSC generation, differentiation, and culturing protocols are needed. Current efforts towards more standardization in stem cell use in research by the International Society for Stem Cell Research aims at making results more comparable and interoperable, eventually facilitating a better and more efficient clinical translation of basic research findings (www.isscr.org/standards-document) accessed on 22 August 2023. This should integrate alongside a shared access to BMP rare disease patient material, which is crucial to facilitate generation of more patient-derived iPSC lines and new differentiation protocol workflows. Ultimately, modelling human diseases with iPSCs will speed up the identification of druggable targets and the identification of new pharmaceutical compounds for treating FOP, PAH, and HHT.

## 5. Conclusions

Since the invention of iPSC technology, FOP, PAH, and HHT research has greatly benefited from cellular human disease models in investigating underlying pathomechanisms and new drug targets. These developments complement conventional approaches, such as use of genetic animal models, which, in some cases, did not gain the anticipated insights. BMP signalling and its cellular consequences are highly complex and diverse in a cell-type specific manner, resulting in fine-tuned canonical SMAD and non-canonical pathways, which crosstalk and interplay with each other and with a variety of other signalling related to ECM, biomechanics, inflammation, hypoxia, and other growth factors. The current aim for several research groups is to mimic this complex, multifaceted cellular microenvironment as closely as technically possible to establish adequate cellular human FOP, PAH, and HHT disease models. Those models will continue to show value in the preclinical validation of promising new compounds. Besides easier access to the already established models, which we summed up here, new iPSC-derived models may be necessary to cater to the needs of increasing omics-based analysis and further exploration of more specific differentiation protocols. It is important to also expand iPSC technology towards BMP-related diseases that are currently not addressed.

## Figures and Tables

**Figure 2 cells-12-02200-f002:**
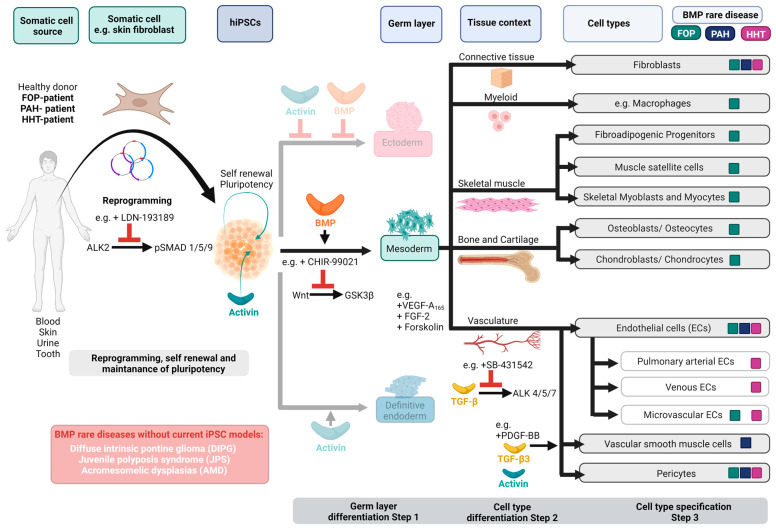
**iPSC generation and cell type differentiation in rare BMP diseases**. So far, iPSC lines have been successfully generated for three rare BMP diseases: Fibrodysplasia ossificans progressiva (FOP), Hereditary pulmonary arterial hypertension (HPAH), and Hereditary haemorrhagic telangiectasia (HHT). Various sources of somatic tissue have been explored for this purpose. Reprogramming efficiency may be affected by the BMP gene mutation, and small molecules, like LDN-193189, that aim to restore normal BMP signalling can be utilized. BMPs are potent inducers of mesoderm. Therefore, a combination of BMP agonists and Wnt antagonists (such as CHIR-99021) is employed to induce mesodermal differentiation. In order to generate vascular cells, ALK4/5/7 inhibition is often necessary in a subsequent step.

**Figure 4 cells-12-02200-f004:**
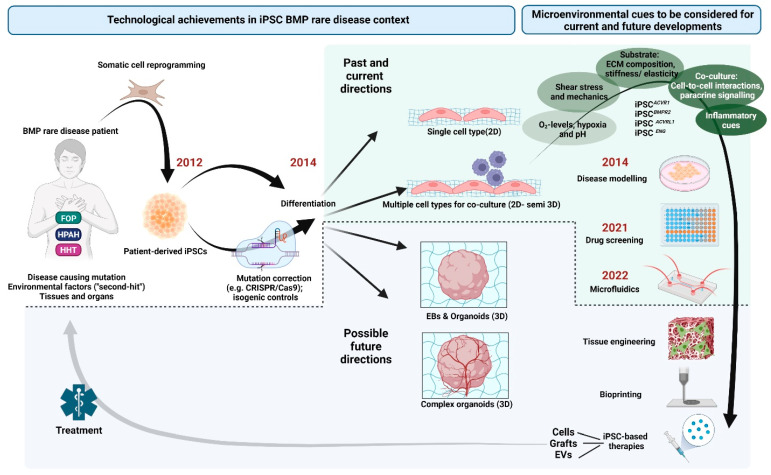
**Current and Future Development of iPSC-based technology in BMP rare diseases**. Since the inception of the first iPSC-based cell models, significant progress has been made in developing more intricate and robust systems that accurately replicate diseased tissues and organs. Of particular interest in monogenic diseases is the utilization of gene editing tools such as CRISPR/Cas9 to create isogenic cell lines. While the first culturing approaches opted for 2 dimensional methods, more complex three dimensional culture conditions, such as embryoid bodies (EBs) and organoids, are emerging. This can be combined with co-culture of different cell types differentiated from the same parental iPSC line. Culturing those cells under conditions that expose them to microenvironmental cues, such as hypoxia, biomechanics, and inflammation, allows for effective and complex disease modelling. These systems have proven to be valuable platforms for drug screening and in-depth mechanistic investigations. Further advances, such as construction of complex cellular grafts or therapeutic extracellular vesicles (EVs) derived from iPSCs, hold the potential to revolutionize tissue engineering and iPSC-based cell therapies, presenting therapeutic alternatives for patients affected by rare BMP diseases.

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
