# Peer review of "Human iPSCs as Model Systems for BMP-Related Rare Diseases"

_cells, 2023, doi:10.3390/cells12172200_

Round 1

Reviewer 1 Report

The authors summarize the current state of literature on iPSC-derived model systems for BMP signaling in rare diseases associated with mutations in the Bone morphogenetic protein (BMP) signaling pathways. However, the authors need to include more specific details. Teh signaling model for BMP receptors, especially on the plasma membrane, regulation of signaling by membrane proteins and endocytosis, Proteins binding to BMP receptors and how these proteins regulate the signaling, non canonical BMP signaling pathways. There are multiple proteins identified to interact with BMP type II and type I receptors that play crucial roles in signaling. For example phosphatases such as PTPN1 and kinases such as CK2 are crucial for regulation. Antagonists and agonists and how they influence BMP signaling, pseudo receptors BAMBI etc must be discussed. Limitations of current research in context of the pathways should be discussed. What questions are open, what proteins should be explored.

Author Response

Reviewer #1

“The authors summarize the current state of literature on iPSC-derived model systems for BMP signaling in rare diseases associated with mutations in the Bone morphogenetic protein (BMP) signaling pathways. However, the authors need to include more specific details. Teh signaling model for BMP receptors, especially on the plasma membrane, regulation of signaling by membrane proteins and endocytosis. Proteins binding to BMP receptors and how these proteins regulate the signaling. non canonical BMP signaling pathways. There are multiple proteins identified to interact with BMP type lI and type I receptors that play crucial roles in signaling. Forexample phosphatases such as PTPN1 and kinases such as CK2 are crucial for regulation. Antagonists and agonists and how they influence BMP signaling. pseudo receptors BAMBI etc must be discussed.”

Authors: We agree with this reviewer that BMP signaling is extremely complex. As such, a myriad of factors are involved in its regulation at multiple levels, from soluble ligands availability to transcriptional and translational output. Furthermore, activation of BMP receptors regulates an intricated network of intracellular effectors and bidirectional crosstalk with other pathways. Given this, our aim in this review is to facilitate the research on BMP signaling to researchers skilled in iPSCs, while promoting the use of iPSCs to BMP experts. Therefore, we have intended very carefully to balance these two major subjects, while keeping the text accessible enough to readers from both fields. Non-canonical signaling was already illustrated in the original version. To satisfy the concerns raised by this reviewer, we have now included references to BAMBI, Neogenin, CRIM1, CK2, PTPN1, LCK and FHIT to illustrate some regulators of ligand-receptor activation. We have added the corresponding explanations and citations in lines 105 to 117. SARA and Endofin are now cited as mediators of TGFBR internalization via endocytosis in lines 142 to 148.

Both authors would like to emphasize that this review aimed at delivering a summary of available information on iPSC technology and BMP related rare diseases. Unfortunately, to keep it comprehensive and readable, we cannot deliver detailed information and review all available knowledge concerning SMAD signaling and non canonical signaling pathways. In that respect, both authors have authored and co-authored  extensive reviews on BMP signaling and BMP receptor associated molecules in the past  (Sánchez-Duffhues G et al. Bone. 2015;80:43-59; Sánchez-Duffhues G, et al. Bone. 2016;93:220-1; Cai J, Pardali E et al. FEBS Lett. 2012;586(14):1993-2002. Gomez-Puerto MC et al. J Pathol. 2019;247(1):9-20; Sanchez-Duffhues G et al., Bone. 2020;138:115472; Ventura F et al., Biomedicines. 2021;9(2). and also with particular focus on BMP receptor interacting proteins and non canonical signaling (Sieber C et al. Cytokine Growth Factor Rev. 2009:343-55; Hiepen C et al. Cells. 2020). Our aim was here to place iPSC- related BMP research into the center of readers attention also acknowledging the aims of the special issue, in which this review shall be published.

R#1: Limitations of current research in context of the pathways should be discussed. What questions are open. what proteins should be explored.

Authors: In the section “Future perspectives”, we have included a paragraph to emphasize the new challenges to investigate BMP signaling, resulting from the establishment of iPSC-based microenvironment resembling platforms, and a potential way how to overcome them. To address the concerns of the reviewer, we have additionally added in lines 855-872 new “pathway specific” points to be considered in future research commenting on SMAD vs. non-SMAD signaling as well as “lateral SMAD” signaling, which we think is important for future research in understanding the pathomechanisms of FOP, PAH and HHT. We have also added a new section “Conclusions” in lines 912- 924 where we comment on current achievements and future aspects to consider including use of iPSC technology for yet underdiscovered BMP-related diseases.

Reviewer 2 Report

 Disturbances in Bone morphogenetic protein (BMP) signalling contribute to onset and development of a number of rare genetic diseases, including fibrodysplasia ossificans progressiva  (FOP), pulmonary arterial hypertension (PAH) and hereditary haemorrhagic telangiectasia (HHT). In this review the current state of literature on iPSC-derived model systems in this field with special emphasis on the access to patient source material and the complications that may come with it. Given the essential role of BMPs during embryonic development and stem cell differentiation, gain- or loss-of-function mutations in BMP signalling pathway may compromise iPSC generation, maintenance and differentiation procedures. In this review they discuss recent developments approaching towards complex culture models aiming to resemble specific tissue microenvironments with multi-faceted cellular inputs, such as cell mechanics and ECM together with organoids, organ-on-chip and microfluidic technologies. 

This review is well written. What has been achieved so far as well as what has yet to be achieved is discussed in detail.There is sufficient description as a review, and it is judged as acceptable.

Author Response

Reviewer #2

“Disturbances in Bone morphogenetic protein (BMP) signaling contribute to onset and development of a number of rare genetic diseases, including fibrodysplasia ossificans progressiva (FOP), pulmonary arterial hypertension (PAH) and hereditary hemorrhagic telangiectasia (HHT). In this review the current state of literature on iPSC-derived model systems in this field with special emphasis on the access to patient source material and the complications that may come with it. Given the essential role of BMPs during embryonic development and stem cell differentiation, gain- or loss-of-function mutations in BMP signalling pathway may compromise iPSC generation, maintenance and differentiation procedures. In this review they discuss recent developments approaching towards complex culture models aiming to resemble specific tissue microenvironments with multi-faceted cellular inputs, such as cell mechanics and

ECM together with organoids, organ-on-chip and microfluidic technologies. This review is well written. What has been achieved so far as well as what has yet to be achieved

is discussed in detail. There is sufficient description as a review, and it is judged as acceptable.”

Authors: We are very thankful for the praise and positive comments from this reviewer. In our revision to address the concerns from Reviewer#1, we have done our best to avoid compromising the strengths of the manuscript highlighted by this reviewer.